



# Key Landscapes for Conservation Land Cover and Change Monitoring Thematic and Validation Datasets for Sub-Saharan Africa

Zoltan Szantoi[1,2], Andreas Brink[1], Andrea Lupi[1], Claudio Mammone[3], Gabriel Jaffrain[4]

[1]European Commission, Joint Research Centre, Directorate for Sustainable Resources, 21027 Ispra, Italy
[2]Department of Geography and Environmental Studies, Stellenbosch University, Stellenbosch 7602, South Africa
[3]e-Geos - an ASI / Telespazio Company, Contrada Terlecchie, 75100, Matera, Italy
[4]IGN FI - Ingénierie Géographique Numérique Française à l'International, 75012 Paris, France

*Correspondence to*: Zoltan Szantoi (zoltan.szantoi@ec.europa.eu)

**Abstract.** Threats to biodiversity pose an enormous challenge for Africa. Mounting social and economic demands on natural resources increasingly threaten key areas for conservation. Effective protection of sites of strategic conservation importance requires timely and highly detailed geospatial monitoring. Larger ecological zones and wildlife corridors warrant monitoring
as well, as these areas have an even higher degree of pressure and habitat loss. To address this, a satellite imagery based monitoring workflow to cover at-risk areas at various details was developed. During the program's first phase, a total of 560,442km$^2$ area in Sub-Saharan Africa was covered, from which 153,665km$^2$ were mapped with 8 land cover classes while 406,776km$^2$ were mapped with up to 32 classes. Satellite imagery was used to generate dense time series data from which thematic land cover maps were derived. Each map and change map were fully verified and validated by an independent team
to achieve our strict data quality requirements. The independent validation datasets for each KLCs are also described and presented here (The complete dataset available at Szantoi et al., 2020A https://doi.pangaea.de/10.1594/PANGAEA.914261, and a demonstration dataset at Szantoi et al., 2020B https://doi.pangaea.de/10.1594/PANGAEA.915849).

## 1 Introduction

Key Landscapes for Conservation (MacKinnon et al., 2015) (KLC) are defined as areas vast enough to sustain large wild
animals (e.g. Big Five game) within functioning biomes that face pressure from various external factors such as poaching, agriculture expansion and urbanization. Land use changes cause loss in both flora and fauna, by altering wild animal movements and eventually decrease their populations (Di Minin et al., 2016; van der Meer, 2018). The livelihood of People and wildlife in Africa that depend on natural resources face increasing pressure from demand by the continent's increasing population, set to reach 2 billion by 2040 (MacKinnon et al., 2015) (Di Minin et al., 2016). The representative location types,
often transboundary, of the KLCs uniquely positions them as benchmarks for their natural resources management to generate



steady income for the local residents while protecting their wildlife (MacKinnon et al., 2015). Benchmarking activities of this kind require highly accurate thematic land cover/change (LCC) map products. Although LCC maps exist for many areas within Africa, the majority of products only cover protected areas with some buffer zones (Szantoi et al., 2016). However, continental and global mapping efforts reported thematic accuracies for such land cover maps between 67%-81%, with lower class

accuracies reported in many cases. (Mora et al., 2014). Differences in legends and methods make these cases difficult to use for monitoring, modelling or change detection studies. In order to use various LC and LCC products together (i.e. modelling, policy making), land cover class definitions should be standardized to avoid discrepancies in thematic class understanding. Not all users (international organizations, national governments, civil societies, researchers) have the capabilities to readjust such maps (Saah et al., 2020). To accommodate diverse user profiles, a common processing scheme is employed. The resulting

datasets can be utilized through various platforms and systems.

This work adopts the Land Cover Classification Scheme of the Food and Agriculture Organization (FAO LCCS, DiGregorio 2005), an internationally approved ISO standard approach. The presented datasets in this paper are produced within the Copernicus High Resolution Hot Spot Monitoring (HSM) activity of the Copernicus Global Land Service. All HSM products feature the same thematic land cover legend and geometric accuracy, and were processed and validated following the same

methodology. In addition, as all Copernicus products, including the HSM data, are free and open to any user with guaranteed long-term maintenance and availability.

Copernicus serves as an operational program where data production takes place on a continuous basis. This paper presents twelve KLC land cover [change] datasets that cover up to 560,442km$^2$ terrestrial land area in Sub-Saharan Africa (SSA) mapped under the first phase (Phase 1) of the HSM activity. The datasets are based on freely available medium spatial

resolution data. Each of the KLCs were individually validated for both present (~2016) and change (~2000) dates. The developed processing chain always consists of preliminary data assessment for availability, pre and post processing as well as fully independent quality verification and validation steps. For the latter, a second dataset called validation data is presented as well.

Several recent studies call for the sharing of product validation datasets (Fritz et al., 2017; Tsendbazar et al., 2018), especially

if a collection received financial support from government grants (Szantoi et al., accepted, 2020C). Accordingly, the validation datasets (LC/LCC) associated with each of the KLCs are also shared.

## 2 Study Area

The provided thematic datasets concentrate on Sub-Saharan Africa, a region on the frontline of natural and human induced changes. The selection of areas were conducted based on present and future pressures envisioned and predicted (MacKinnon

et al., 2015). In this first phase (Phase 1) 12 large areas totalling 560442km$^2$ in SSA were selected, mapped and validated (Figure 1). These areas cover various ecosystems and generally reside in transboundary regions (Table 1, Figure 1).



**Table 1 Mapped Key Landscapes for Conservation (KLC) within Phase 1.  Mapping detail refers to the employed classification scheme – Dichotomous (D) and Modular (M); see it in the Data collection and mapping guidelines section.**

| **KLC** (MacKinnon et al., 2015) | **Code** | **Mapping detail** | **Ecoregion** (Dinerstein et al., 2017) | **Country** | **Area (km²)** |
|---|---|---|---|---|---|
| Takamanda | CAF01 | M | Cameroon Highlands forests, Cross-Sanaga-Bioko coastal forests, Guinean and Northern Congolian Forest-Savanna | Nigeria, Cameroon | 79534 |
| Greater Virunga | CAF02 | M | Albertine Rift montane forests Victoria Basin forest-savanna | DRC, Uganda, Rwanda | 39062 |
| Manovo-Gounda-St Floris-Bamingui | CAF06 | M | East Sudanian savanna | Central African Republic, Chad | 96965 |
| Salonga | CAF07 | D | Central Congolian lowland forests | DRC | 66625 |
| Upemba | CAF11 | M | Central Zambezian wet miombo woodlands | DRC | 47318 |
| Lomami | CAF15 | M | Central Congolian lowland forests | DRC | 30924 |
| Mbam Djerem | CAF16 | D | Northern Congolian Forest-Savanna Northwest Congolian lowland forests | Cameroon | 11510 |
| Yangambi* | CAF99 | M | Northeast Congolian lowland forests | DRC | 7276 |
| Great Limpopo | SAF02 | M | Zambezian mopane woodlands Limpopo lowveld | Mozambique, South Africa, Zimbabwe | 65475 |
| North and South Luangwa | SAF14/ SAF15 | D | Dry miombo woodlands Central Zambezian wet miombo woodlands | Zambia | 34880 |
| Comoe-Mole | WAF05 | D | West Sudanian savanna Guinean forest-savanna | Ivory Coast, Ghana | 40648 |
| Tai-Sapo | WAF10 | M | Western Guinean lowland forests | Ivory Coast, Liberia | 40219 |
| **Area total** | | | | | **560442** |

* - it is not included in MacKinnon et al. (2015) list. DRC - Democratic Republic of the Congo

**Figure 1 Spatial distribution of the Key Landscapes for Conservation Phase 1 areas.**



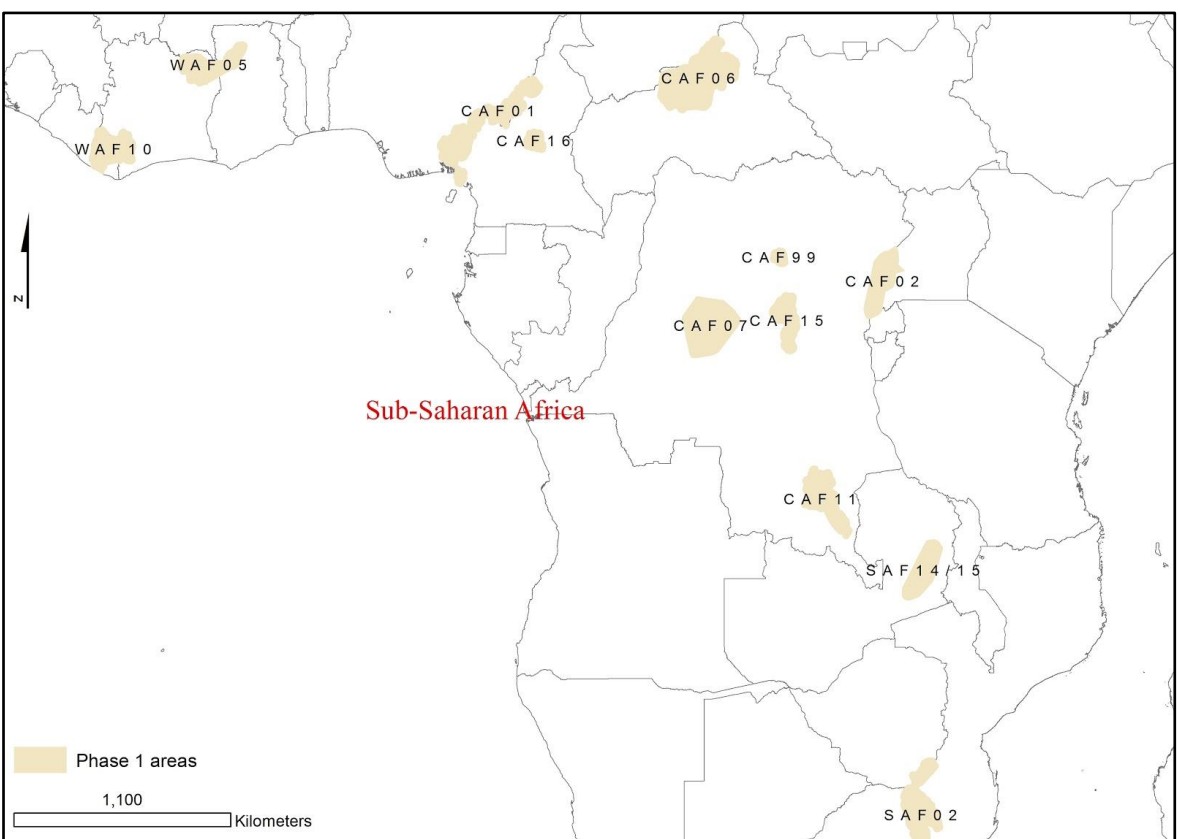

# 3 Data and Method

## 3.1 Thematic dataset production

### 3.1.1 Data collection and mapping guidelines

Landsat TM, ETM+ and OLI imagery at Level1TP processing level were used in the production of the Phase 1 land cover and change maps. The Level1TP data was further corrected for atmospheric conditions to produce surface reflectance products for

the classification phase. The atmospheric correction module was implemented based on the 6S as a direct radiative transfer model (Masek et al., 2006). The Shuttle Radar Topography Mission (30m or 90m) Digital Elevation Model was used to estimate the target height and slope, as well as correct the surface sun incidence angles to perform an optional topographic correction. The Aerosol Optical Thickness (AOT) was estimated directly from either Landsat or Sentinel-2 data (Hagolle et al., 2015). Based on the area's meteo-climatic conditions (climate profile and precipitation patterns), season specific satellite

image data were selected for each KLC (Table 1). Due to data scarcity for many areas, especially for the change maps (year 2000), imagery was collected for a target year ± 3 years. In extreme cases, (±) 5 years were allowed, or until four cloud free





observations per pixel for the specified date were reached. The cloud and shadow masking procedure was based on the FMASK algorithm (Zhu et al., 2015).

### 3.1.2 Land cover classification system

All thematic maps were produced either at *Dichotomous*, or at both Dichotomous and *Modular* levels within the Land Cover Classification System (LCCS) developed by the Food and Agriculture Organization of the United Nations and the United Nations Environment Programme (Di Gregorio, 2005). The LCCS (ISO 19144-2) is a comprehensive hierarchical classification system, which enables comparison of land cover classes regardless of geographic location or mapping date and scale (Di Gregorio, 2005). At the *Dichotomous* level, the system distinguishes eight major LC classes. At the *Modular* level,

thirty-two LC classes were used (Table 2).

**Table 2 Dichotomous and Modular thematic land cover/use classes.**

| Dichotomous level | Mapcode | Modular level | Mapcode |
|---|---|---|---|
| Cultivated and Managed Terrestrial Area (A11) | 3 | continuous large to medium sized field (>2 ha) of tree crop cover: plantation | 31 |
| | | continuous small sized field (<2 ha) of tree crop cover: plantation | 32 |
| | | continuous large to medium sized field (>2 ha) of tree crop cover: orchard | 33 |
| | | continuous small sized field (<2 ha) of tree crop cover: orchard | 34 |
| | | continuous large to medium sized field (>2 ha) of shrub crop | 55 |
| | | continuous small sized field (<2 ha) of shrub crop | 56 |
| | | continuous large to medium sized field (>2 ha) of herbaceous crop | 59 |
| | | continuous small sized field (<2 ha) of herbaceous crop | 60 |
| Natural and Semi-Natural Primarily Terrestrial Vegetation (A12) | 4 | continuous closed (>70-60) trees | 77 |
| | | continuous open general (70-60)-(20-10)% trees | 78 |
| | | continuous closed to open (100-40)% shrubs | 112 |
| | | continuous open (40 - (20-10)%) shrubs | 116 |
| | | continuous closed to open (100-40)% herbaceous vegetation | 148 |
| | | continuous open (40 - (20-10)%) herbaceous vegetation | 152 |
| Cultivated Aquatic or Regularly Flooded Area (A23) | 6 | continuous large to medium sized field (>2 ha) of woody crops | 155 |
| | | continuous small sized field (<2 ha) of woody crops | 156 |
| | | continuous large to medium sized field (>2 ha) of graminoid crops | 159 |



| | | continuous small sized field (<2 ha) of graminoid crops | 160 |
|---|---|---|---|
| Natural And Semi-Natural Aquatic or Regularly Flooded Vegetation (A24) | 7 | closed (>70-60)% trees | 165 |
| | | open general (70-60)-(20-10)% trees | 166 |
| | | closed to open (100-40)% shrubs | 171 |
| | | very open (40 - (20-10)%) shrubs | 175 |
| | | closed to open (100-40)% herbaceous vegetation | 178 |
| | | very open (40 - (20-10)%) herbaceous vegetation | 182 |
| Artificial Surfaces and Associated Area (B15) | 10 | built up area | 184 |
| | | non built up area | 185 |
| Bare Area (B16) | 11 | Bare area | 11 |
| Artificial Waterbodies, Snow and Ice (B27) | 13 | artificial waterbodies (flowing) | 186 |
| | | artificial waterbodies (standing) | 187 |
| Natural Waterbodies, Snow and Ice (B28) | 14 | natural waterbodies (flowing) | 190 |
| | | natural waterbodies (standing) | 191 |
| | | snow | 192 |
| | | ice | 193 |

### 3.1.3 Automatic classification

Based on the pre-selected imagery data, Dense Multitemporal Timeseries (DMT) based vegetation indices were generated to

reduce data dimensionality and enhance the signal of the surface target. The DMT for each KLCs were based on the pre-processed and geometrically coregistered data, forming a geospatial datacube (Strobl et al., 2017). In addition, three vegetation indices were calculated to aid the separation of terrestrial vs. aquatic (NDFI), vegetated vs. barren (SAVI), and evergreen vs. deciduous vegetation areas (NBR).

The indices are (per Landsat spectral bands):

*Normalized Difference Flooding Index (NDFI)*   $NDFI = \frac{(RED-SWIR2)}{(RED+SWIR2)}$   (1)

*Soil Adjusted Vegetation Index (SAVI)*   $SAVI = \frac{1.5 \times (NIR-RED)}{(NIR+RED+0.5)}$   (2)

*Normalized Burn Ratio (NBR)*   $NBR = \frac{(NIR-SWIR2)}{(NIR+SWIR2)}$   (3)

All the pre-processed data (spectral bands and the DMT based indices) were fed into the Support Vector Machine supervised

classification model. The Support Vector Machine classifier can handle data with high dimensionality and performs well with mapping heterogeneous areas, including vegetation community types (Szantoi et al., 2013). To produce the thematic maps, the Minimum Mapping Unit concept used by Szantoi et al. (2016) was employed. Individual pixels (with corresponding land cover class information) were assigned into objects, where the minimum size of an object was set at 0.5-5 hectares, as a compromise between technical feasibility (pixel size) and the general size of the observable features (various land cover classes). Still,

classification errors (omission and commission of various classes) and false alarms (for land cover change) arose due to the data availability (cloud cover, no data) and the seasonal behaviour of the land cover (e.g. rapid foliage change). To correct



these errors, expert human image interpretation skills and knowledge that improved the outputs from the automated process were used.

## 3.2 Validation dataset production

The validation datasets (Table 3, Figure 2) were individually created for each KLCs. The validation datasets (points) were generated using a stratified random sampling procedure. This assured a sufficient estimation for all land cover and land cover change classes according to their frequency of occurrence. The following formula (Gallaun et al., 2015) was used to determine the minimum number of validation points (per class per KLC):

$$n_c = \frac{p_c(1-p_c)}{\sigma_c^2}, c = 1, \dots, L \qquad (4)$$

$n_c$ number of sampling units for class c

$p_c$ estimated error rate for class c

$\sigma_c$ accepted standard error of the error of commission for class c

$L$ number of classes

In cases where classes covered smaller areas in total, additional sampling units were allocated according to the Neyman optimal allocation in order to minimize the variance of the estimator of the overall accuracy for the total sample size [n] (Gallaun et al., 2015; Stehman, 2012):

$$n_c = \frac{nN_c\sigma_c}{\sum_{k=1}^{L} N_k\sigma_k} \qquad (5)$$

$n_c$ sample size for class c

$N_c$ population size for class c

$\sigma_c$ estimated error rate for class c

$L$ number of classes

$N_k$ population size for class k

$\sigma_k$ estimated error rate for class k

At least two independent data analysts (blind and plausibility interpretation process) evaluated all accuracy points. Eventually, some points were excluded from the accuracy statistics due to an error/disagreement during the evaluation procedure (Table 3

- "Number of points LC/LCC"). The *blind* process attempt to interpret all validation points was based on available ancillary data (i.e. higher resolution imagery), without direct comparison to the generated LC/LCC maps. The *plausibility* process



reviewed every point whose the blind interpretation did not match the corresponding LC/LCC value (disagreement between the LC/LCC data and the blind interpretation). After this review, the final validation reference is established.

**Table 3 Validation dataset attributes**

| KLC Code | Mapping detail | Number of LC classes | Number of LCC classes | Number of points LC/LCC |
|----------|----------------|----------------------|-----------------------|-------------------------|
| CAF01    | M              | 26                   | 12                    | 3849                    |
| CAF02    | M              | 26                   | 18                    | 4465                    |
| CAF06    | M              | 19                   | 13                    | 4151                    |
| CAF07    | D              | 5                    | 3                     | 1364                    |
| CAF11    | M              | 23                   | 15                    | 3785                    |
| CAF15    | M              | 17                   | 9                     | 3687                    |
| CAF16    | D              | 7                    | 2                     | 1254                    |
| CAF99    | M              | 17                   | 14                    | 2727                    |
| SAF02    | M              | 26                   | 19                    | 3367                    |
| SAF14/15 | D              | 6                    | 3                     | 1335                    |
| WAF05    | D              | 8                    | 3                     | 1264                    |
| WAF10    | M              | 22                   | 12                    | 4423                    |





**Figure 2 Spatial distribution of the validation datasets within each Key Landscapes for Conservation areas.**





## 150 4. Assessment - Data Quality

**Technical Validation**

*Spatial, temporal and logical consistency* was assessed by an independent procedure from the producer to determine the products positional accuracy, the validity of data with respect to time (seasonality), and the logical consistency of the data (topology, attribution and logical relationships). A Qualitative-systematic accuracy assessment was also performed wall-to-

155 wall through a systematic visual examination for a) global thematic assessment b) expected size of polygons (MMU), c) seasonal effects and d) spatial patterns (i.e. following correct edges).

The quantitative accuracy assessment (i.e. validation) results are shown in Table 4 (overall accuracies), and in the Appendix (thematic class accuracies per KLC, Appendix A). Generally, the program aimed at a minimum of 85% overall accuracy for each product (KLC) and a minimum of 75% thematic accuracy (Producer's and User's) for each class within each KLC. The

160 land cover change (LCC) accuracy should be >72%. In exceptional cases, the thematic accuracies might be lower than the threshold due to the difficulty to discriminate a particular class in a certain KLC. Figure 3 shows the final LC and LCC products classified at the dichotomous LCCS level while Figures 4A and 4B show the final LC and LCC products classified at the modular LCCS level.

**Table 4 Achieved overall accuracies for land cover mapping in (%)**

| KLC Code | Land cover map [200X] | Reference date | Land cover map [201X] | Reference date |
|---|---|---|---|---|
| CAF01 | 94.31 | 2000 | 92.26 | 2016 |
| CAF02 | 91.93 | 2001 | 90.09 | 2015 |
| CAF06 | 87.82 | 2003 | 85.72 | 2015 |
| CAF07 | 99.40 | 2000 | 99.60 | 2016 |
| CAF11 | 96.10 | 2000 | 95.27 | 2016 |
| CAF15 | 99.10 | 2000 | 99.10 | 2016 |
| CAF16 | 99.10 | 2000 | 98.90 | 2016 |
| CAF99 | 98.12 | 2000 | 98.51 | 2016 |
| SAF02 | 93.32 | 2002 | 92.8 | 2016 |
| SAF14/15 | 97.70 | 2000 | 97.70 | 2015 |
| WAF05 | 97.10 | 2000 | 96.40 | 2015 |
| WAF10 | 98.43 | 2001 | 98.78 | 2016 |



**Figure 3 Key Landscapes for Conservation - Dichotomous classification level. The boundaries (black polygons) represent protected areas (IUCN category I-IV) within the KLCs. Both, land cover and land cover change, are presented for each KLC.**



**Figure 4A Key Landscapes for Conservation - Modular classification level. The boundaries (black polygons) represent protected areas (IUCN category I-IV) within the KLCs. Both, land cover and land cover change, are presented for each KLC.**





**Figure 4B Key Landscapes for Conservation - Modular classification level. The boundaries (black polygons) represent protected areas (IUCN category I-IV) within the KLCs. Both land cover and land cover change are presented for each KLC.**

## 5. Discussion

There is a direct relationship between population growth, agricultural expansion, energy demand and pressure on land. With the current state of development, population increase and economic growth, a large portion of the Sub-Saharan population





depend on the remaining natural resources to meet their food and energy needs (Brink et al., 2012). The demands of social and economic growth require additional land, often on the expense of previously untouched areas. Often areas under protection (i.e. National Parks), that remain well-preserved (see Figures 3 and 4AB), have regions in close proximity under tremendous pressure. Such areas (many times transboundary ones) need very accurate monitoring and base maps, which are provided through this work; especially as areas shared between and/or among countries are many times not mapped with a common legend, if mapped at all. The presented KLC datasets can be used for continuous land cover/use monitoring, evaluation of management practices/effectiveness, endowment for scientific counsel, habitat modelling, information dissemination and capacity building in their corresponding countries and to manage natural resources such as forests, soil, biodiversity, ecosystem services and agriculture (Tolessa et al., 2017). Furthermore, regional climate change, biogeochemical and hydrologic models are currently capable  of using high resolution LC data for predictions in general (Nissan et al., 2019) and spatially focused (i.e. Africa) (Sylla et al., 2016; Vondou and Haensler, 2017).

The validation datasets are independently collected and verified through a robust procedure. Validation datasets can then be used for additional land cover mapping, creating spectral libraries, and for the validation of other local, regional and global datasets. It is important that various land cover products can be used or compared against one another regardless of their geographic locations. Here, twelve land cover maps for different areas in Sub-Saharan Africa where quality land cover products are missing (Marshall et al., 2017) were introduced. These products come with land cover change information as well, generally dating back to year 2000 (±3 years). All data were produced using the unified Land Cover Classification System. The LCCS's modular level can be applied to local scales through its very detailed classes (here 32).

## 5.1 Drivers of change

Geist and Lambin (2002) describe the human driving forces of land-cover changes as an interlinking of three key variables: expansion of agriculture, extraction of wood, and development of infrastructure. The main land cover dynamic in Sub-Saharan Africa can be explained by the first two variables, where agriculture expansion is further subdivided into shifting cultivation, permanent cultivation, and cattle ranching, and wood extraction is subdivided into commercial wood extraction (clear-cutting, selective harvesting), fuelwood extraction, pole wood extraction, and charcoal production. Although the driving force behind the clearing of natural vegetation has traditionally been attributed mainly to the expansion of new agricultural land areas (including investments in large-scale commercial agriculture) (Brink and Eva, 2009), firewood extraction and charcoal production are also key factors in forest, woodland and shrub land degradation throughout the region. This land cover dynamic is not just a by-product of greater forces such as logging for timber and agricultural expansion, but stems from a specific need to satisfy energy demand (European Commission, 2018); in fact, in Sub-Saharan Africa, the main use of extracted wood is for energy production (Kebede et al., 2010). Although the region possesses a huge diversity of energy sources such as oil, gas, coal, uranium, and hydropower, the local infrastructure and use of these commercial energy sources are very limited. Traditional sources of energy in the form of firewood and charcoal account for over 75% of the total energy use in the region (Kebede et al., 2010). Efforts to meet the population and economic demands in sub-Saharan Africa while preserving

biodiversity and ecosystem functioning require informed decision-making. The Copernicus Global Land Service, in particular the High-Resolution Hot Spot Monitoring component, present a unique opportunity for such information gathering.

## 5.2 Sources of errors

As the applied LCCS allows very detailed hierarchical classification, some classes can be difficult to distinguish from each other. This is especially true in Africa's vast and very heterogeneous landscapes where agricultural land use is mainly smallholder based (i.e. very small plots), while shifting cultivation is mostly due to the lack of fertilizers and weak soil, leading to land abandonment. Landscapes are generally not composed of clearly fragmented and well identifiable cover formation. In this region, landscapes usually form a continuum of various cover (vegetation) formations that might include different layers

of tree, shrub and herbaceous. These variations combined with differences in vegetation density (open vs. closed) and heights makes class assignments challenging. Moreover, some specific agriculture classes distinguish even the cultivation type, e.g. differentiating between fruit tree plantations from tree plantations for timber. Thus, the discrimination of such classes are very difficult and might introduce classification errors.

Apart from the land cover classification, errors could also be introduced due to climate-induced variability, such as leaf

phenology where deciduous vegetation might appear bare during a dry period (season).

At a more general level, difficulties in identifying between aquatic or regularly flooded surfaces and terrestrial areas have been observed in certain KLCs, especially when flooded periods are short.

## 5.3 Datasets current and future use

The C-HSM datasets have been widely used by policy makers (African and European partners) to help identify areas prone to

change due to human activities. For example, COFED (Support Unit for the [DRC] National Authorizing Officer of the European Development Fund) the EEAS (European External Action Service) of the DRC manage an envelope of EUR120m, allocated for five protected areas in the DRC (Virunga, Garamba, Salonga, Upemba and the Yangambi biosphere), where they use the C-HSM products for planning and for investment strategies (i.e. hydropower). Another example comes from West Africa, where NGOs (i.e. Wild Chimpanzee Foundation),  public-benefit enterprises (i.e. German Society for International

Cooperation - GIZ) as well as national authorities (i.e. l'Office Ivoirien des Parcs et Réserves - OIPR) use the data to identify areas under pressure for agriculture (cocoa, oil palm, rubber, coconut) and human-wildlife conflicts in Cote d'Ivoire, Ghana and Liberia.

## 6. Data Availability

The data are provided in a shapefile (*.shp) format, polygon geometry for the land cover and change datasets and point

geometry for the validation datasets. The presented data is in the World Geodetic System 1984 Geographic Coordinate System

(GCS) (EPSG:4326) and its datum (EPSG:6326). The validation data, beside using the same GCS, also have the Africa Albers Equal Area Conic (EPSG:102022) projected coordinate system.

Each of the 12 areas is described by two vector layers, a Land Cover (LC) layer and a Land Cover Change (LCC) layer. The LC layer is a wall-to-wall map, covering the entire Area of Interest (AOI). The LC temporal reference for the project is the year 2016, but for each area the actual "mapping year" is noted in the file name (i.e. CAF01_2016), and it generally refers to the year in which the largest number of satellite images were used for the classification. The LCC layer provides a partial coverage of the AOI, as it contains only the areas (polygons) where thematic change occurred compared to the LC layer. The LCC temporal reference is the year 2000 (+/- 3 years), noted in the file name (i.e. CAF01_2000).

Each LC and LCC shapefiles comes with its corresponding attribute table, where two or three attributes are present: [mapcode_A] - dichotomous class, [mapcode_B] - modular class, [name_A] - corresponding dichotomous classnames (KLCs classified only at the dichotomous level, [name_B] - corresponding modular classname.

Validation points dataset:

Each of the 12 areas has been quantitatively validated using a spatially specific point dataset. These datasets were generated through the method described in point 3.2, and each point was used to verify the correctness of the LC/LCC maps. The corresponding data in the attribute table are: LC - [plaus201X] and LCC - [plaus200X]. Both [plaus201X] and [plaus200X] attributes refer to the most detailed classification level attributes [mapcode_A or mapcode_B] present in the LC and LCC

datasets (shapefiles).

Demonstration dataset:

The demonstration dataset contains one land cover thematic map (Greater Virunga KLC – CAF02, 2015, modular level, three attributes: [mapcode_A], [mapcode_B] and [name_B]) and the corresponding validation dataset ([plaus2015]).

The naming of all attributes follow the same structure in all data. Please see the details in the Appendix Information and Supplementary Information section.

Besides archiving the complete and the demonstration datasets at PANGEA (www.pangea.eu) with corresponding Digital

Object Identifiers (complete: Szantoi et al., 2020B, https://doi.pangaea.de/10.1594/PANGAEA.914261, demonstration: Szantoi et al., 2020C, https://doi.pangaea.de/10.1594/PANGAEA.915849), the Copernicus Hot-Spot website (https://land.copernicus.eu/global/hsm) provides open access to all the land cover/change data presented in this article as well as technical reports and on the fly statistics.



## 7. Conclusions and Outlook

The C-HSM service component is part of the Copernicus Global Land Service, which produces near real time biophysical variables at medium scale, globally. In contrast, the C-HSM activity is an on demand component addressing specific user requests in the field of sustainable management of natural resources. The products presented here provide the first set of standardized land cover and land cover change datasets for 12 KLCs with their corresponding validation datasets (200X, and 201X) in Sub-Saharan Africa. The geographic distribution covers the tropical and subtropical regions of West, Central and

South-Eastern Africa. The next release will also include countries in the Caribbean and Pacific areas of the ACP region and some areas beyond these regions may be mapped depending on user demands. The most recent land cover change will be reassessed for selected already-mapped KLC's in order to generate longer-term time series land cover dynamics information. While this is not done systematically, but on specific customer requests, the C-HSM service encourages stakeholder cooperation and provides capacity building workshops where the service teaches not only how to use the data and the web

information system, but also how to easily assess recent land cover change using Sentinel 2 image mosaics.

Finally, the service has a high degree of confidence that the data presented here (and the next phase) are of highest quality, reaching regularly above 90% overall accuracy. This is guaranteed by a rigorous and independent production-validation mechanism and feedback loop, which does not stop until the required overall, and per-class accuracy levels are reached.

Following the general European Commission's Copernicus open access policy, the data is distributed free to any user through

a dedicated website (https://land.copernicus.eu/global/hsm). This interactive online information system allows access to browse, analyse and download the data, including the accuracy assessment information.

## Appendix Information

Appendix A contains the thematic class accuracies for each KLC, both land cover and land cover maps.

CLASS_A - Corresponding class (see Table 2 'Dichotomous map code') - OR

CLASS_B - Corresponding class (see Table 2 'Modular map code')

PA - Producer's accuracy

UA - User's accuracy

NoRP - number of reference points



## Appendix A.

**Thematic class accuracies per KLC***

*Accuracy parameters are in percent, classes with less than 15 samples were not included in the overall accuracy calculation.

| CAF01 | | | | | | | |
|-------|-----|-----|------|---------|-----|-----|------|
| 2000 | | | | 2016 | | | |
| CLASS_B | PA | UA | NoRP | CLASS_B | PA | UA | NoRP |
| 3 | 96.3 | 93.9 | 903 | 11 | 98.1 | 96.4 | 64 |
| 4 | 90.4 | 96.6 | 1061 | 31 | 94.7 | 89.3 | 283 |
| 6 | 100 | 90 | 46 | 32 | 86.5 | 90.4 | 61 |
| 7 | 95.8 | 93.5 | 206 | 33 | 77 | 93.5 | 7 |
| 11 | 98.2 | 96.4 | 63 | 34 | 74 | 43.3 | 12 |
| 13 | 100 | 93.4 | 57 | 55 | 92.4 | 100 | 62 |
| 14 | 95.4 | 91.2 | 159 | 56 | 99.5 | 96.7 | 91 |
| 77 | 97.5 | 96.5 | 654 | 59 | 89.4 | 82.4 | 45 |
| 78 | 91.8 | 84.9 | 429 | 60 | 90.3 | 90.7 | 401 |
| 165 | 96.7 | 89.5 | 106 | 77 | 97.7 | 96.2 | 584 |
| 166 | 69.3 | 83.6 | 15 | 78 | 90.6 | 85.3 | 414 |
| 184 | 99.7 | 94.1 | 100 | 112 | 81.6 | 92.8 | 458 |
| 185 | 89.3 | 89.6 | 44 | 116 | 92 | 87.7 | 270 |
| | | | | 148 | 87 | 92.8 | 225 |
| | | | | 152 | 84.4 | 99.5 | 25 |
| | | | | 160 | 100 | 89.8 | 46 |
| | | | | 165 | 96.6 | 89.3 | 108 |
| | | | | 166 | 73.9 | 84.7 | 15 |
| | | | | 171 | 94.3 | 94.1 | 103 |
| | | | | 175 | 69.6 | 61.1 | 4 |
| | | | | 178 | 99.9 | 92 | 97 |
| | | | | 184 | 99.7 | 93.9 | 172 |
| | | | | 185 | 97 | 89.1 | 83 |
| | | | | 187 | 95.3 | 96.7 | 61 |
| | | | | 190 | 95.7 | 90.9 | 97 |





| 191 | 100 | 95 | 61 |
|---|---|---|---|
| | | | 325 |

| CAF02 | | | | | | | |
|---|---|---|---|---|---|---|---|
| 2001 | | | | 2015 | | | |
| CLASS_B | PA | UA | NoRP | CLASS_B | PA | UA | NoRP |
| 3 | 95.4 | 95.7 | 1523 | 11 | 99.9 | 98.8 | 130 |
| 4 | 86.5 | 91.7 | 1054 | 31 | 64.9 | 88.3 | 150 |
| 6 | 0 | 0 | 1 | 32 | 89.5 | 91 | 287 |
| 7 | 87.4 | 84.3 | 362 | 33 | 0 | 0 | 1 |
| 11 | 88.9 | 92.9 | 94 | 34 | 88.1 | 95.5 | 123 |
| 14 | 99.6 | 99.7 | 370 | 55 | 87.5 | 60.3 | 9 |
| 77 | 93.2 | 87 | 686 | 56 | 92.9 | 88.3 | 558 |
| 78 | 65.3 | 67.7 | 160 | 59 | 69.8 | 93.6 | 27 |
| 165 | 50.5 | 38.3 | 8 | 60 | 89.5 | 93.9 | 569 |
| 166 | 86.9 | 85.3 | 16 | 77 | 96.5 | 91.6 | 544 |
| 184 | 87 | 89.8 | 122 | 78 | 61.2 | 74.7 | 153 |
| 185 | 97.7 | 81.1 | 39 | 112 | 82.4 | 76.8 | 237 |
| 192 | 100 | 100 | 30 | 116 | 90.9 | 85 | 269 |
| | | | | 148 | 86.1 | 92 | 322 |
| | | | | 152 | 94 | 99.3 | 3 |
| | | | | 160 | 0 | 0 | 1 |
| | | | | 165 | 77.8 | 37.6 | 7 |
| | | | | 166 | 56.2 | 85.1 | 16 |
| | | | | 171 | 82.3 | 84.8 | 176 |
| | | | | 175 | 63.8 | 56.9 | 15 |
| | | | | 178 | 84.7 | 72.3 | 214 |
| | | | | 182 | 100 | 69.2 | 1 |
| | | | | 184 | 88.9 | 98.1 | 213 |
| | | | | 185 | 89.6 | 58 | 44 |
| | | | | 190 | 88.3 | 99.2 | 80 |
| | | | | 191 | 100 | 99.6 | 286 |
| | | | | 192 | 100 | 100 | 30 |





| CAF06 | | | | | | | |
|-------|-----|-----|------|-------|-----|-----|------|
| 2003 | | | | 2015 | | | |
| CLASS_B | PA | UA | NoRP | CLASS_B | PA | UA | NoRP |
| 3 | 82.6 | 91.5 | 236 | 55 | 100 | 100 | 47 |
| 4 | 88.9 | 93.3 | 1882 | 60 | 80.5 | 89.1 | 199 |
| 7 | 98.3 | 76.1 | 422 | 77 | 83.4 | 92.2 | 656 |
| 14 | 99.4 | 90.5 | 103 | 78 | 85.8 | 77.2 | 738 |
| 77 | 83.5 | 92.1 | 680 | 112 | 85.7 | 90.7 | 1427 |
| 78 | 85.8 | 77.2 | 749 | 116 | 83.2 | 84.3 | 280 |
| 184 | 91.9 | 89.9 | 73 | 148 | 90.5 | 91.5 | 127 |
| | | | | 171 | 96.4 | 64.3 | 113 |
| | | | | 175 | 96.5 | 70 | 123 |
| | | | | 178 | 87.8 | 88.4 | 173 |
| | | | | 184 | 93.4 | 91 | 128 |
| | | | | 190 | 99.4 | 90 | 71 |
| | | | | 191 | 100 | 99.8 | 32 |

| CAF07 | | | | | | | |
|-------|-----|-----|------|-------|-----|-----|------|
| 2000 | | | | 2016 | | | |
| CLASS_A | PA | UA | NoRP | CLASS_A | PA | UA | NoRP |
| 3 | 96 | 89.4 | 120 | 3 | 99.7 | 96.5 | 127 |
| 4 | 99.4 | 99.9 | 847 | 4 | 99.5 | 100 | 836 |
| 7 | 100 | 97.6 | 255 | 7 | 100 | 97.6 | 255 |
| 10 | 100 | 89.7 | 61 | 10 | 100 | 94.2 | 65 |
| 14 | 100 | 99.2 | 81 | 14 | 100 | 99.2 | 81 |

| CAF11 | | | | | | | |
|-------|-----|-----|------|-------|-----|-----|------|
| 2000 | | | | 2016 | | | |
| CLASS_B | PA | UA | NoRP | CLASS_B | PA | UA | NoRP |
| 3 | 98.7 | 92.8 | 320 | 11 | 100 | 100 | 30 |
| 4 | 99.3 | 93.8 | 1125 | 32 | 100 | 100 | 26 |
| 6 | 100 | 14.4 | 1 | 34 | 0 | 0 | 0 |





| | | | | | | | |
|---|---|---|---|---|---|---|---|
| 7 | 96.9 | 99.2 | 618 | 56 | 69.9 | 100 | 2 |
| 11 | 100 | 96.7 | 29 | 59 | 92.4 | 99.1 | 75 |
| 14 | 98.7 | 99.9 | 278 | 60 | 97.3 | 97.1 | 334 |
| 77 | 94.5 | 95.6 | 539 | 77 | 94.6 | 95.2 | 488 |
| 78 | 92.6 | 97.7 | 652 | 78 | 92.4 | 97.1 | 584 |
| 165 | 79.4 | 96.3 | 77 | 112 | 96.8 | 86.9 | 405 |
| 166 | 98.7 | 99.2 | 48 | 116 | 97.7 | 94.3 | 284 |
| 184 | 100 | 95.8 | 83 | 148 | 98.5 | 97.1 | 321 |
| 185 | 100 | 95.4 | 15 | 152 | 0 | 0 | 0 |
| | | | | 160 | 100 | 100 | 3 |
| | | | | 165 | 79.1 | 96.2 | 76 |
| | | | | 166 | 96.9 | 99.2 | 47 |
| | | | | 171 | 75 | 92.7 | 77 |
| | | | | 175 | 56.8 | 98.6 | 74 |
| | | | | 178 | 97.9 | 98 | 411 |
| | | | | 182 | 95 | 95 | 20 |
| | | | | 184 | 100 | 98.9 | 161 |
| | | | | 185 | 100 | 100 | 75 |
| | | | | 190 | 87.9 | 98.2 | 89 |
| | | | | 191 | 99.8 | 100 | 203 |

| CAF15 | | | | | | | |
|---|---|---|---|---|---|---|---|
| 2000 | | | | 2016 | | | |
| CLASS_B | PA | UA | NoRP | CLASS_B | PA | UA | NoRP |
| 3 | 100 | 82.8 | 80 | 77 | 99.7 | 99.5 | 1936 |
| 4 | 98.3 | 95.8 | 546 | 78 | 94.1 | 91.9 | 257 |
| 7 | 78.5 | 94.2 | 108 | 112 | 93.1 | 92.7 | 379 |
| 14 | 98.2 | 96.9 | 97 | 116 | 0 | 0 | 3 |
| 77 | 99.7 | 99.5 | 2048 | 148 | 98.9 | 97.2 | 306 |
| 78 | 91.9 | 92.4 | 303 | 152 | 100 | 86.4 | 57 |
| 165 | 94.1 | 98.7 | 348 | 165 | 94.1 | 98.8 | 300 |
| 166 | 100 | 81.4 | 72 | 166 | 100 | 81.2 | 63 |




| 184 | 98.3 | 95.8 | 85 | 171 | 74.2 | 88.7 | 41 |
| | | | | 175 | 0 | 0 | 1 |
| | | | | 178 | 83.5 | 95.8 | 69 |
| | | | | 184 | 100 | 99.7 | 178 |
| | | | | 190 | 98.2 | 96.9 | 97 |

| CAF16 | | | | | | | |
|---|---|---|---|---|---|---|---|
| 2000 | | | | 2016 | | | |
| CLASS_A | PA | UA | NoRP | CLASS_A | PA | UA | NoRP |
| 3 | 96.8 | 72.5 | 93 | 3 | 88.3 | 84.6 | 142 |
| 4 | 99.5 | 99.7 | 848 | 4 | 99.3 | 99.5 | 761 |
| 7 | 86.4 | 82.6 | 94 | 7 | 85.7 | 82.6 | 94 |
| 10 | 96.2 | 98.1 | 55 | 10 | 97.3 | 98.7 | 94 |
| 13 | 100 | 98.7 | 75 | 13 | 100 | 94.7 | 75 |
| 14 | 96.1 | 94.9 | 73 | 14 | 96.1 | 94.9 | 73 |

| CAF99 | | | | | | | |
|---|---|---|---|---|---|---|---|
| 2000 | | | | 2016 | | | |
| CLASS_B | PA | UA | NoRP | CLASS_B | PA | UA | NoRP |
| 3 | 91.6 | 98.9 | 431 | 31 | 91.6 | 99.8 | 267 |
| 4 | 92.4 | 92.1 | 417 | 32 | 94.5 | 100 | 69 |
| 7 | 100 | 97.8 | 231 | 56 | 100 | 99.5 | 76 |
| 14 | 100 | 100 | 175 | 59 | 100 | 9.5 | 4 |
| 77 | 99 | 99.2 | 905 | 60 | 91.9 | 96.5 | 125 |
| 78 | 93.6 | 85.1 | 210 | 77 | 99.6 | 99.2 | 732 |
| 165 | 97.8 | 97.9 | 246 | 78 | 79.1 | 91.5 | 156 |
| 166 | 100 | 88.7 | 40 | 112 | 96.1 | 95.9 | 341 |
| 184 | 99.4 | 88.3 | 72 | 148 | 98.7 | 96.9 | 168 |
| | | | | 165 | 97.8 | 97.5 | 240 |
| | | | | 166 | 100 | 89.2 | 42 |
| | | | | 171 | 100 | 100 | 102 |
| | | | | 175 | 0 | 0 | 3 |



| 178 | 100 | 91.6 | 77 |
| 184 | 100 | 95.9 | 150 |
| 185 | 100 | 100 | 2 |
| 190 | 100 | 100 | 113 |
| 191 | 100 | 100 | 60 |

| SAF02 | | | | | | | |
|---|---|---|---|---|---|---|---|
| 2002 | | | | 2016 | | | |
| CLASS_B | PA | UA | NoRP | CLASS_B | PA | UA | NoRP |
| 3 | 93.9 | 94.9 | 705 | 11 | 98.3 | 100 | 3 |
| 4 | 96.1 | 96 | 1425 | 31 | 100 | 86.1 | 66 |
| 6 | 100 | 67 | 1 | 33 | 93.8 | 88.1 | 104 |
| 7 | 94.7 | 91.3 | 170 | 34 | 98.1 | 76.8 | 140 |
| 11 | 100 | 100 | 2 | 55 | 84.1 | 40.3 | 30 |
| 13 | 91.9 | 98.3 | 76 | 56 | 55 | 100 | 3 |
| 14 | 91.5 | 92.7 | 146 | 59 | 96.6 | 95 | 185 |
| 77 | 84.7 | 75.8 | 204 | 60 | 91.7 | 92.7 | 165 |
| 78 | 81.2 | 85.1 | 392 | 77 | 85 | 74.3 | 154 |
| 165 | 11.4 | 84.1 | 7 | 78 | 79 | 87.2 | 400 |
| 166 | 90.8 | 98.6 | 17 | 112 | 96.8 | 94.7 | 880 |
| 184 | 92.7 | 92.6 | 142 | 116 | 90.9 | 96.2 | 284 |
| 185 | 100 | 94.7 | 67 | 148 | 77.6 | 94.2 | 122 |
| | | | | 152 | 85.1 | 87.6 | 108 |
| | | | | 160 | 100 | 100 | 3 |
| | | | | 165 | 0 | 0 | 4 |
| | | | | 166 | 91.6 | 100 | 13 |
| | | | | 171 | 98.5 | 90.8 | 100 |
| | | | | 175 | 78.9 | 78 | 35 |
| | | | | 178 | 92.6 | 93.9 | 42 |
| | | | | 182 | 100 | 50 | 2 |
| | | | | 184 | 94.8 | 97.3 | 211 |
| | | | | 185 | 100 | 95.1 | 93 |



| 187 | 95.9 | 98.4 | 83 |
| 190 | 96.6 | 99.2 | 100 |
| 191 | 83.7 | 87.3 | 24 |

| SAF14/15 | | | | | | | |
|---|---|---|---|---|---|---|---|
| 2000 | | | | 2015 | | | |
| CLASS_A | PA | UA | NoRP | CLASS_A | PA | UA | NoRP |
| 3 | 91 | 94.8 | 215 | 3 | 95.9 | 95.2 | 301 |
| 4 | 98.7 | 99.2 | 845 | 4 | 98.6 | 99.2 | 756 |
| 7 | 93.4 | 84.2 | 73 | 7 | 93.5 | 88.6 | 74 |
| 10 | 96 | 81.6 | 67 | 10 | 96.8 | 84.6 | 77 |
| 11 | 100 | 100 | 42 | 11 | 100 | 100 | 42 |
| 14 | 85.1 | 87.4 | 85 | 14 | 85.2 | 87.4 | 85 |

| WAF05 | | | | | | | |
|---|---|---|---|---|---|---|---|
| 2000 | | | | 2015 | | | |
| CLASS_A | PA | UA | NoRP | CLASS_A | PA | UA | NoRP |
| 3 | 77.2 | 97.6 | 217 | 3 | 83.2 | 99.3 | 310 |
| 4 | 99.5 | 97.4 | 735 | 4 | 99.6 | 96.1 | 583 |
| 6 | 0 | 0 | 0 | 6 | 0 | 0 | 0 |
| 7 | 98.5 | 77.9 | 26 | 7 | 81.6 | 77.9 | 26 |
| 10 | 95.2 | 93.2 | 77 | 10 | 100 | 98.1 | 138 |
| 11 | 100 | 100 | 57 | 11 | 100 | 100 | 57 |
| 13 | 100 | 96 | 72 | 13 | 100 | 93.3 | 70 |
| 14 | 100 | 100 | 74 | 14 | 100 | 100 | 74 |

| WAF10 | | | | | | | |
|---|---|---|---|---|---|---|---|
| 2001 | | | | 2016 | | | |
| CLASS_B | PA | UA | NoRP | CLASS_B | PA | UA | NoRP |
| 3 | 96 | 98.6 | 1518 | 11 | 100 | 100 | 32 |
| 4 | 94.5 | 100 | 151 | 31 | 94.2 | 99.3 | 275 |
| 6 | 66.9 | 100 | 44 | 32 | 87.3 | 100 | 3 |



| | | | | | | |
|---|---|---|---|---|---|---|
| 7 | 99.2 | 93.8 | 79 | 33 | 100 | 50 | 1 |
| 11 | 100 | 100 | 32 | 34 | 100 | 92.7 | 22 |
| 13 | 100 | 100 | 109 | 55 | 0 | 0 | 13 |
| 14 | 99.3 | 100 | 94 | 56 | 99.5 | 97.8 | 1153 |
| 77 | 99.5 | 98.8 | 2017 | 59 | 0 | 0 | 2 |
| 78 | 93.3 | 91.5 | 215 | 60 | 95 | 98.3 | 327 |
| 165 | 100 | 96.8 | 43 | 77 | 99.5 | 99.6 | 1695 |
| 166 | 0 | 0 | 0 | 78 | 93.4 | 90.8 | 189 |
| 184 | 99.3 | 98.9 | 83 | 112 | 98.8 | 95.7 | 32 |
| 185 | 0 | 0 | 0 | 116 | 100 | 100 | 1 |
| | | | | 148 | 98.6 | 99.9 | 100 |
| | | | | 152 | 0 | 0 | 1 |
| | | | | 160 | 68.1 | 100 | 50 |
| | | | | 165 | 88.9 | 96.8 | 44 |
| | | | | 166 | 0 | 0 | 1 |
| | | | | 171 | 100 | 96.9 | 59 |
| | | | | 178 | 99 | 86.7 | 20 |
| | | | | 184 | 93.5 | 100 | 159 |
| | | | | 185 | 100 | 42.1 | 2 |
| | | | | 187 | 100 | 100 | 109 |
| | | | | 190 | 98.9 | 100 | 95 |
| | | | | 191 | 0 | 0 | 0 |

**Supplement.** The supplement related to this article is available online at: XXX (please see STable 1).

STable 1 contains references to the data which have been used to create the land cover/change maps for each KLC.

Sensor:

L8 - Landsat 8 imagery

L7 - Landsat 7 imagery

L5 - Landsat 5 imagery

Date (acquisition of the imagery): YYYYMMDD - year - month - day format

Path/Row of the Imagery used: Worldwide Reference System for Landsat data (i.e. scene location on the globe)



**Author contribution**: ZSZ and ABB designed the work. CM and GJ implemented the workflows. ZSZ and ABB wrote the paper. ZSZ, ABB, AL and GJ revised the paper.

**Acknowledgements.** The development of the thematic maps have been made possible thanks to the effort of ITHACA (Information Technology for Humanitarian Assistance, Cooperation and Action) and Telespazio - a LEONARDO And THALES company; and their quality evaluations by IGNFI (France), Joanneum Research (Austria), EOXPLORE (Germany), GISBOX (Romania), Space4environment (Luxembourg), ONFI (France) and LuxSpace (Luxembourg). The authors also thank Mr. A. McKinnon (EC/JRC) for proofreading the paper.

**Competing interests.** The authors declare that they have no conflict of interest.

**Review statement.** This paper was edited by XY and reviewed by X anonymous referees.

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
