# Peer review of "Key Landscapes for Conservation Land Cover and Change Monitoring, Thematic and Validation Datasets for Sub-Saharan Africa"

_Earth System Science Data, 2020_

## Referee Comment (RC1) · Anonymous Referee #1 · 15 Jun 2020

This manuscript describes the classification of land cover and land cover change on a selection of sites of defined important to biodiversity in Africa. Spatial assessment of land cover and land cover change is, as the authors note, a valuable tool in efforts to conserve biodiversity. The manuscript describes actions taken to produce accurate land cover maps using a repeatable legend for multiple sites. As such it makes a useful contribution to an expanding array of tools and data sources. Importantly, the data set and validation data are made available for users. The open availability of such data is to be welcomed.

[Figure]

2020.

---

## Referee Comment (RC2) · Anonymous Referee #2 · 18 Jun 2020

The article is of a high quality and addresses a very important issue of data scarcity, and taking into account an immensely important component of sustainable development in Sub-Saharan Africa.

The methodology provided looks very sound and easy to follow. The map validation and accuracy assessment process was robust resulting in very high accuracy, given the complex and heterogeneous nature of the landscape in the study areas.

The data is of good quality and easy to visualize at the Copernicus Hot-Spot website. However, in its present form the data is too large for download in countries with limited internet bandwidth. Although the Copernicus Hot-Spot website provides open access

to all the data presented in the article, the value added could be enhanced, by ensuring the data is packaged in manageable form for download by resource managers and decision makers. Additionally, making the validation data set open and accessible will facilitate replication of the mapping process by technicians in the pilot countries

---

## Referee Comment (RC3) · Anonymous Referee #3 · 19 Jun 2020

The monitoring framework presented by Szantoi et al. is most welcomed, especially for African countries that may not readily have the necessary resources and tools, and / or have access to the required resources for developing such a monitoring framework. Such a monitoring framework is invaluable, and I anticipate interest and use by many organizations globally. The datasets are easily downloadable over a high speed internet. However, I suspect this may not be true for many African countries / organizations that have limited internet access. Perhaps the authors could recommend / give thought to alternatives for easing data accessibility. The authors should also comment on how countries with limited expertise can improve their automated outputs considering that classification errors and false alarms are inevitable.

[Figure]

Please also note the supplement to this comment:
https://essd.copernicus.org/preprints/essd-2020-77/essd-2020-77-RC3-
supplement.pdf

———————————————————

[Figure]

**Supplement:**

**General comments**

The monitoring framework presented by Szantoi et al. is most welcomed, especially for African countries that may not readily have the necessary resources and tools, and / or have access to the required resources for developing such a monitoring framework. Such a monitoring framework is invaluable, and I anticipate interest and use by many organizations globally. This research is a significant contribution to the broader discipline of land cover mapping and monitoring.

**Specific comments**

The manuscript is well written, coherent, and readily understandable, albeit several grammar and typographical transgressions.

Consider including a diagram illustrating the overall workflow.

Include relevant details regarding the change analysis. This is significant in the context of the high classification results achieved.

**Technical corrections**

Title: check grammar; perhaps a colon should be placed after "Monitoring"?

Formatting of values and units in the Abstract.

Text, formatting (e.g. references), and typographical errors in the Introduction.

Inconsistency in formatting of values, e.g. L60.

Figure 1; indicate country names for easy reference.

Confusion regarding "200X", "201X", "plaus200X", and "plaus201X". Brief explain / clarify.

Define all acronyms the first time used, e.g. MMU, C-HSM.

Section 5.3: "i.e." should rather be "e.g."?

---

## Author Comment (AC1) · 6 Jul 2020

Dear Reviewer,

Thank you for emphasizing the importance of our datasets.

Best regards Zoltan Szantoi on behalf of all the Co-Authors

---

## Author Response (AR1)

5 **Anonymous Referee #1**

This manuscript describes the classification of land cover and land cover change on

a selection of sites of defined important to biodiversity in Africa. Spatial assessment

of land cover and land cover change is, as the authors note, a valuable tool in efforts

10 to conserve biodiversity. The manuscript describes actions taken to produce accurate

land cover maps using a repeatable legend for multiple sites. As such it makes a useful

contribution to an expanding array of tools and data sources. Importantly, the data set

and validation data are made available for users. The open availability of such data is

to be welcomed.

**Dear Reviewer#1**

Thank you for the review. We appreciate your comments.
* * *
Interactive comment on "Key Landscapes for

20 Conservation Land Cover and Change Monitoring

Thematic and Validation Datasets for Sub-Saharan

Africa" by Zoltan Szantoi et al.

**Anonymous Referee #2**

25 The article is of a high quality and addresses a very important issue of data scarcity,

and taking into account an immensely important component of sustainable development

in Sub-Saharan Africa.

The methodology provided looks very sound and easy to follow. The map validation

30 and accuracy assessment process was robust resulting in very high accuracy, given

the complex and heterogeneous nature of the landscape in the study areas.

The data is of good quality and easy to visualize at the Copernicus Hot-Spot website.

However, in its present form the data is too large for download in countries with limited

internet bandwidth. Although the Copernicus Hot-Spot website provides open access to all the data presented in the article,

35 the value added could be enhanced, by ensuring

the data is packaged in manageable form for download by resource managers and

decision makers. Additionally, making the validation data set open and accessible will

facilitate replication of the mapping process by technicians in the pilot countries

40 **Dear Reviewer #2**

Thank you for your notes. Based on your recommendations, we generated a source dataset at https://doi.pangaea.de/10.1594/PANGAEA.914261. Within this source dataset, users can download the complete package (354MB) containing all the KLCs, as well as there is an option to download each KLC site separately. The links to download the individual KLC datasets are provided at the https://doi.pangaea.de/10.1594/PANGAEA.914261 website under "*Source*

45 *data set:*". Each of the datasets contain the corresponding land cover and land cover change maps, QGIS legends files, and the validation dataset.

We also added both the land cover/change and the validation datasets to the Copernicus Hot-Spot website (https://land.copernicus.eu/global/hsm), where users can visualize and/or download them directly from the selected KLC panel.

50 All datasets (land cover/change and validation) are fully open.
* * *
Interactive comment on "Key Landscapes for

Conservation Land Cover and Change Monitoring

Thematic and Validation Datasets for Sub-Saharan

55 Africa" by Zoltan Szantoi et al.

**Anonymous Referee #3**

The monitoring framework presented by Szantoi et al. is most welcomed, especially

for African countries that may not readily have the necessary resources and tools,

60 and / or have access to the required resources for developing such a monitoring

framework. Such a monitoring framework is invaluable, and I anticipate interest and

use by many organizations globally. The datasets are easily downloadable over

a high speed internet. However, I suspect this may not be true for many African

countries / organizations that have limited internet access. Perhaps the authors

65 could recommend / give thought to alternatives for easing data accessibility. The

authors should also comment on how countries with limited expertise can improve their

automated outputs considering that classification errors and false alarms are inevitable. Please also note the supplement to this comment:

https://essd.copernicus.org/preprints/essd-2020-77/essd-2020-77-RC3-

70   Supplement.pdf

**Dear Reviewer#3**

We appreciate your comments and recommendations. Please see below our response to each of your comments and questions.

75   **"Perhaps the authors could recommend / give thought to alternatives for easing data accessibility."**

We received a similar comment from Reviewer #2 as well - please see our response there too. Based on both notes, we (1) generated smaller data packages (i.e. each KLC can be downloaded individually) from https://doi.pangaea.de/10.1594/PANGAEA.914261, where the full data package can also be downloaded. As of now, the

80   individual data packages range from as little as 1.4MB (Mbam-Djerem, CAF16) to 133MB (Takamanda, CAF01), based on their mapping details and areas and (2) we also published all land cover and land cover change data, as well as the validation datasets at the Copernicus Hot-Spot website (https://land.copernicus.eu/global/hsm) for quick visualization with option to download the individual KLCs.

85   **"The authors should also comment on how countries with limited expertise can improve their automated outputs considering that classification errors and false alarms are inevitable."**

Through this work we provide very high quality products which can be used directly as base maps (e.g. Yangambi KLC) and for policy decisions (e.g. all KLCs in the Democratic Republic in the Congo - by the European External Action Service).

90   However, if a local agency or government has a map producing processing chain, they can use the presented land cover maps to compare and/or evaluate their outputs for change detection or use the validation datasets for training purposes. In this way, given the high accuracy of our products, classification errors (omission and commission of various classes) and false alarms (land cover change) can be filtered and corrected on the locally produced map product or based on the detected errors, their classification processing chains can be updated.

95

Moreover, if an even more detailed land cover/use product is needed, our products can be used as an existing base to narrow down where certain land cover/use might be present. For example, in the case of non-industrial cocoa plantations detection, our legend does not have such detail. However, our product allows users to narrow down to A.) Cultivated and Managed Terrestrial Area (A11), and from there to B.) Continuous Small Sized Field of Shrub Crop (Mapcode 56). Within these areas,

100   users can employ very high resolution imagery to discriminate the above mentioned cocoa plantations.

We added a specific sentence to our Conclusions and Outlook section in the main text:

"Here, we provide very high-quality products, which can be used directly as base maps and for policy decisions, as well as for comparison and/or evaluation of other land cover products or the implementation of validation datasets for training/validation

105 purposes."

**General comments - from the supplement**

**The monitoring framework presented by Szantoi et al. is most welcomed, especially for African countries that may not readily have the necessary resources and tools, and / or have access to the required resources for developing such a**

110 **monitoring framework. Such a monitoring framework is invaluable, and I anticipate interest and use by many organizations globally. This research is a significant contribution to the broader discipline of land cover mapping and monitoring.**

**Specific comments**

115 **The manuscript is well written, coherent, and readily understandable, albeit several grammar and typographical transgressions.**

Thank you - we double checked the entire manuscript for grammar and typographical transgressions.

**Consider including a diagram illustrating the overall workflow.**

120 We added it as a new figure (Figure 2), titled "Overall production workflow".

[Figure]

**Include relevant details regarding the change analysis. This is significant in the context of the high classification results achieved.**

125

We added a new section - 3.1.4 Land cover change detection

Land cover change was interpreted as a categorical change in which a particular land cover was replaced by another land cover. As an example of conversion, the change of Cultivated and Managed Terrestrial Areas (A11) into a Natural and Semi-Natural Terrestrial Vegetation (A12) or a Cultivated and Managed Terrestrial Areas (A11) into Artificial Surfaces and Associated Areas (B15) can be mentioned. The basic condition for LC changes identification was the detection of changes in spectral reflectance within specific image bands of the employed satellite imagery, but such changes were further evidenced by other interpretation parameters such as shape and texture patterns. In regards to our methodology, images acquired in two or more different timeframes were used in the identification process. Furthermore, land cover changes were characterised by those changes that have longer than yearly and/or seasonal periodicity (dry/wet season). Urban sprawl, tree plantations (large or small) to replace herbaceous crops (large or small), tree covers (closed or open) or the creation of a new water reservoir undergo long-term changes that classify as actual LCCs. In our workflow, the LCC process followed the same image pre-processing steps as the LC method, and an independent classification (similarly to the LC procedure) of the past date was performed. Finally, the LC and the LCC products were compared and change polygons were extracted. As with the LC product, the visual refinement was an important step to produce accurate LCC polygons.

**Technical corrections**

**Title: check grammar; perhaps a colon should be placed after "Monitoring"?**

Done. We checked the entire manuscript for grammar.

**Formatting of values and units in the Abstract.**

Done. Changed to full numbers (e.g. 345670km$^2$ instead of 345,670km$^2$) throughout the manuscript.

**Text, formatting (e.g. references), and typographical errors in the Introduction.**

Done. We double checked the entire manuscript for errors.

**Inconsistency in formatting of values, e.g. L60.**

Done. Changed to full numbers (e.g. 345670km$^2$ instead of 345,670km$^2$) throughout the manuscript.

**Figure 1; indicate country names for easy reference.**

Done. We updated Figure 1 accordingly.

[Figure]

160 **Confusion regarding "200X", "201X", "plaus200X", and "plaus201X". Brief explain / clarify.**

Done. We added a short explanation to Table 4 - "*[200X] and [201X] refer to the year the map represent; the exact year is in the "Reference date" columns" and to the Data Availability section - "The plaus201X and plaus200X refer to the year the validation sets represent, as these can be different among KLCs; the exact year is always noted in the columns' names (e.g. plaus2000, plaus2016)."

165

**Define all acronyms the first time used, e.g. MMU, C-HSM.**

Done. We corrected and updated the acronyms.

**Section 5.3: "i.e." should rather be "e.g."?**

170 Done. We changed "i.e." to "e.g."
* * *

[revised manuscript text omitted]

---

## Author Response (AR2)

essd-2020-77: Key Landscapes for Conservation Land Cover and Change Monitoring Thematic and Validation Datasets for Sub-Saharan Africa

Zoltan Szantoi, Andreas Brink, Andrea Lupi, Claudio Mannone, and Gabriel Jaffrain

**Dear Dr. Carlson,**

Thank you for all your help throughout the process. We really appreciate it.

Based on your recommendations (Aug 23) –

We generated small "teaser" datasets for the KLC areas which are larger than 20MB in terms of data size.

Thus, we have 5 additional "teaser" products

1. Szantoi, Zoltan; Brink, Andreas; Lupi, Andrea; Mammone, Claudio; Jaffrain, Gabriel (2020): Teaser - Manovo-Gounda-St Floris-Bamingui Key Landscape for Conservation Land Cover and Validation Datasets (2015). PANGAEA, https://doi.pangaea.de/10.1594/PANGAEA.922846
2. Szantoi, Zoltan; Brink, Andreas; Lupi, Andrea; Mammone, Claudio; Jaffrain, Gabriel (2020): Teaser - Tai-Sapo Key Landscape for Conservation Land Cover and Validation Datasets (2016). PANGAEA, https://doi.pangaea.de/10.1594/PANGAEA.922848
3. Szantoi, Zoltan; Brink, Andreas; Lupi, Andrea; Mammone, Claudio; Jaffrain, Gabriel (2020): Teaser - Takamanda Key Landscape for Conservation Land Cover and Validation Datasets (2016). PANGAEA, https://doi.pangaea.de/10.1594/PANGAEA.922849
4. Szantoi, Zoltan; Brink, Andreas; Lupi, Andrea; Mammone, Claudio; Jaffrain, Gabriel (2020): Teaser - The Great Limpopo Key Landscape for Conservation Land Cover and Validation Datasets (2016). PANGAEA, https://doi.pangaea.de/10.1594/PANGAEA.922851
5. Szantoi, Zoltan; Brink, Andreas; Lupi, Andrea; Mammone, Claudio; Jaffrain, Gabriel (2020): Teaser - Upemba Key Landscape for Conservation Land Cover and Validation Datasets (2016). PANGAEA, https://doi.pangaea.de/10.1594/PANGAEA.922856

All these teaser sets are less than 20MB each, and contain a small subset of their relevant KLC area including validation data. They also have their individual DOIs, but also part of the "source dataset" at:

Szantoi, Zoltan; Brink, Andreas; Lupi, Andrea; Mammone, Claudio; Jaffrain, Gabriel (2020): Land Cover and Change Thematic and Validation Datasets for Sub-Saharan Africa. PANGAEA, https://doi.org/10.1594/PANGAEA.914261

We also defined ACP at line 326:

"The next release will also include countries in the Caribbean and Pacific areas of the ACP region (African, Caribbean and Pacific Group of States, www.acp.int) some areas beyond these regions may be mapped depending on user demands."

Again, thank you for the opportunity!

Best regards

Zoltan Szantoi, on behalf of the co-authors.

Comments to the Author:
Nice description of a useful data product. Thank you for using ESSD.

In regards the file size limitations issue mentioned by two reviewers, even the files by individual sites (as provided by authors) run to 130 MB for e.g. Takamanda KLC. Do the authors consider that file size still prohibitive for users working from limited bandwidth? Could the authors consider a 'teaser' product, with all masks and processing, at limited spatial resolution or coverage or with limited temporal extent? For other ESSD descriptions of large data products, a teaser (e.g. of a reduced subset of the full product) allows users to download / view / test / sample the new product before committing to or arranging bandwidth for a full larger download. The authors will know best what they might choose to show off their product albeit in a package of 20 MB or so. If plausible, a teaser can increase interest and use.

Line 326, ACP not defined.